# Combining lipoic acid to methylene blue reduces the Warburg effect in CHO cells: From TCA cycle activation to enhancing monoclonal antibody production

Léa Montégut[1], Pablo César Martínez-Basilio[1], Jorgelindo da Veiga Moreira[1], Laurent Schwartz[2], Mario Jolicoeur[1]*

1 Department of Chemical Engineering, Research Laboratory in Applied Metabolic Engineering, École Polytechnique de Montréal, Montréal, Québec, Canada, 2 Assistance Publique des Hôpitaux de Paris, Paris, France

* mario.jolicoeur@polymtl.ca

**Data Availability Statement:** All relevant data are within the manuscript and its Supporting Information files.

## Abstract

The Warburg effect, a hallmark of cancer, has recently been identified as a metabolic limitation of Chinese Hamster Ovary (CHO) cells, the primary platform for the production of monoclonal antibodies (mAb). Metabolic engineering approaches, including genetic modifications and feeding strategies, have been attempted to impose the metabolic prevalence of respiration over aerobic glycolysis. Their main objective lies in decreasing lactate production while improving energy efficiency. Although yielding promising increases in productivity, such strategies require long development phases and alter entangled metabolic pathways which singular roles remain unclear. We propose to apply drugs used for the metabolic therapy of cancer to target the Warburg effect at different levels, on CHO cells. The use of α-lipoic acid, a pyruvate dehydrogenase activator, replenished the Krebs cycle through increased anaplerosis but resulted in mitochondrial saturation. The electron shuttle function of a second drug, methylene blue, enhanced the mitochondrial capacity. It pulled on anaplerotic pathways while reducing stress signals and resulted in a 24% increase of the maximum mAb production. Finally, the combination of both drugs proved to be promising for stimulating Krebs cycle activity and mitochondrial respiration. Therefore, drugs used in metabolic therapy are valuable candidates to understand and improve the metabolic limitations of CHO-based bioproduction.

## Introduction

Chinese Hamster Ovary (CHO) cells are considered to be the primary platform for the production of monoclonal antibodies (mAb) and other complex biopharmaceuticals [1]. Research pertaining to improving cell viability, mAb quality, productivity and reproducibility, is still increasing on this topic [2]. Indeed, studies seeking to identify metabolic factors that are limiting cell productivity are only now surfacing [3]. A particular phenotype called the *Warburg effect* is gaining attention since most CHO cell lines exhibit this highly glycolytic metabolism

**Funding:** MJ received the NSERC (https://www. nserc-crsng.gc.ca/) Discovery Grant #RGPIN-2019-05050. The funders had no role in study design, data collection and analysis, decision to publish, or preparation of the manuscript.

**Competing interests:** The authors have declared that no competing interests exist.

despite the presence of oxygen (i.e. aerobic glycolysis) [4, 5]. Due to the Warburg effect, over 75% of pyruvate, the glycolytic end-product, has been reported to be converted into lactate [6]. With such an important carbon flux lost as lactate, the Warburg effect phenomenon clearly limits mAb production [7, 8]. The regulation mechanisms underlying aerobic glycolysis are still poorly understood but they are known to be involved in the limitation of pyruvate progression to the tricarboxylic acid (TCA) cycle and in energetic and redox balances [9, 10].

For the past 30 years, numerous studies have looked at reducing lactate production, aiming to improve culture performances. Restriction of glucose uptake [8, 11, 12] or its replacement [13–15], were found to be conducive for increasing productivity, but hindered cellular growth and prompted doubts on the capacity of mAb glycosylation in such starved cells [16, 17]. Genetic strategies were also tested to control the expression of endogenous or recombinant enzymes for restricting glucose uptake [18, 19], preventing lactate secretion [20] or directly enhancing TCA cycle fluxes [21–24]. However, genetic modifications are sensitive to genome variability and instability and give varying results among cell lines [25].

An alternate strategy to limit the Warburg effect consists of adding biochemical effectors to manipulate specific enzyme activity. For instance, copper ion, a cofactor of many enzymes known to act on mitochondrial targets such as cytochrome c [26] was confirmed to lead to lactate reuptake, TCA activation and increased productivity in CHOs [27–29]. More recently, dichloroacetate (DCA), an effector of pyruvate dehydrogenase kinase (PDHK), was tested in fed-batch CHO cultures. By down-regulating PDHK, this drug is known to increase the activity of pyruvate dehydrogenase (PDH), an enzyme in charge of the entry of pyruvate in the mitochondria [30, 31]. DCA showed to enhance CHO cell viability as well as mAb production with time [32]. With only few such studies published to date, to the best of our knowledge, this is a promising approach that is emerging to understand and manipulate metabolic regulation.

Although the aerobic glycolysis phenotype has been identified in cancer cells since the 1920's [33], it is only since 2011 that Otto Warburg's definition of deregulated cellular energetics was included as part of the "hallmarks of cancer" [34]. This novel approach led to studies on the metabolic therapy of cancer at pre-clinical and clinical levels, testing drugs known to modulate the activity of enzymes that can maximize mitochondrial fluxes [35–39]. In this work, metabolic similarities of CHOs with cancer cells guided the selection of potential drug candidates, among which α-lipoic acid (α-LA), acting at the glycolysis/TCA interface, and methylene blue (MB), enhancing respiratory pathways, were tested. α-LA promotes the entry of pyruvate in the mitochondria by PDHK inactivation [40], and interacts with many other TCA enzymes as well as acting as an anti-oxidant [35, 41]. Effects of α-LA were compared to those of DCA, a compound reported to have similar effects in CHOs [32]. MB, a synthetic dye first prepared by Heinrich Caro in 1876, showed to promote respiration in cancer cells [42], neurons [43, 44] and heart cells [45]. It increases the mitochondrial activity by stimulating the redox exchanges at the mitochondrial membrane [43, 46], thus stimulating proton turnover rate. Our results confirm strategies that limit the Warburg effect and increase mAb production.

## Materials and methods

The ethics committee of the École Polytechnique de Montréal has approved this research under the reference BIO-05/06-01.

### Cell line and medium

The recombinant CHO-DXB11 cell line stably producing the EG2-hFc chimeric monoclonal antibody [47] was kindly provided by Dr. Yves Durocher from the National Research Council

(Montreal, Quebec, Canada). Cells were cultured in SFM4CHO serum-free medium (HyClone, ref. SH305480.2) supplemented with 4 mM glutamine (Gibco, ref. 25030164) and 0.05 mg/mL dextran sulfate (Sigma, ref. D8906). Cells were passaged thrice weekly until reproducible growth curves were achieved, before being put in batch cultures and submitted to drug treatments.

**Culture and drug treatments.** All cultures were seeded at $2.0 \times 10^5$ cells/mL and grown in batch mode, in a humidified incubator at 37˚C and 5% $CO_2$ under gentle agitation (120 rpm). Drugs were added at inoculation (t = 0 h) and cultures were monitored for up to 120 h or until the viability dropped below 90%. Methylene blue (MB) (Laboratoire Mat, ref. BS0110) and sodium dichloroacetate (DCA) (Sigma, ref. 347795) were dissolved directly in the medium. Due to a poor water solubility, alpha-lipoic acid (α-LA) (Sigma, ref. T1395) was dissolved in ethanol and further diluted in culture medium, with a final ethanol concentration ≤ 0.1%. The condition with 0.1% ethanol alone (i.e. control + vehicle) in culture medium was tested to assess the influence of the vehicle on cell metabolism. All supplemented media were sterilized by filtration through a 0.22 μm filter prior to inoculation.

For drug concentration studies, cells were cultured in six-well non-tissue culture treated plates with a final working volume of 3.3 mL and three replicates per condition were prepared. The effect of α- LA was assessed at 10 μM, 20 μM, 50 μM, 100 μM, 200 μM and 500 μM; while MB was used at 10 nM, 100 nM, 500 nM, 1 μM and 10 μM. These concentration ranges were based on those found in literature for α- LA [48–50] and MB [43, 51]. Then, in-depth metabolic assessed as a positive control, at a concentration of 5 mM as suggested in a recent study on CHOs [32]. All cultures were performed in triplicate except for the control, which was replicated in parallel to each experiment (n = 7).

**Cell count and extracellular metabolites measurements.** Cell count and viability were assessed with a hemocytometer according to the trypan blue exclusion method. Volumes of 0.2 mL (6-well plates) or 0.5 mL (shake flasks) were taken every 24 h. 50 μL were used for cell counting and the remaining volume was centrifuged at 200 $g$ for five minutes. Supernatants were frozen (-80˚C) for further analysis: concentrations of glucose, lactate, glutamine and glutamate in the supernatant were assessed with an enzyme-based biochemistry analyzer (YSI 2700 Select, YSI Life Sciences Inc.).

**Respirometry assays.** Measurements of cells specific oxygen consumption rates ($q_{O2}$) were performed as described previously [52]. Briefly, 3 mL of cell suspension was taken daily from each shake flask, counted and put in a 10 mL dissolved gas analysis glass syringe (Hamilton, ref. 81620). If necessary, a larger volume was harvested, centrifuged and resuspended in 3 mL to reach a required minimum of $3.0 \times 10^6$ cells. After addition of small cross-shaped stirring bar, the syringe was sealed hermetically by using the oxygen probe as a plunger to remove the gas phase, and immersed in a 37˚C water bath. Dissolved oxygen concentration was recorded for at least five minutes after the establishment of a steady consumption rate; the final $O_2$ concentration always exceeding 40% air saturation. The cell suspension was then re-oxygenated and a second measurement was performed, ~10 minutes after the addition of the ATP-synthase inhibitor oligomycin A (Sigma, ref. 75351) to a final concentration of 1 μM, to discriminate the respective contributions of oxidative phosphorylation (OxPhos) and mitochondrial proton leak to the global cell respiration. Finally, a second cell count was performed after the complete assay to account for eventual growth and mortality.

**Flow cytometry analysis of mitochondrial membrane potential and ROS production.** For each fluorescent dye, samples of $5 \times 10^5$ live cells were taken daily from each flask and centrifuged 5 min at 200 $g$. Cells were resuspended in 300 μL of culture medium containing 5 μM of MitoSOX (ROS; Invitrogen, ref. M36008) or 10 μg/mL of Rhodamine123 (membrane potential; Invitrogen, ref. R302), incubated in the dark for 30 min, then washed once or twice,

respectively, for 15 min in 300 μL of phosphate-buffered saline (PBS). FACS measurements were performed on 20,000 cells (FACS Canto II, BD). Both MitoSOX and Rhodamine123 were excited at 488 nm, and emissions were collected at 585 ± 21 nm and 530 ± 15 nm, respectively. Data analysis was performed using the FlowJo software (Tree Star Inc.).

**ELISA quantification of mAb production.** The chimeric EG2-hFc monoclonal antibody (mAb) titers were determined by "sandwich" enzyme-linked immunosorbent assay (ELISA). A 96-well high-bind microplate (Corning, ref. 3369) was incubated overnight with goat anti-human Fc-specific antibodies (Sigma, ref. I2136) diluted 1:30,000 in PBS. Wells were washed three times with PBS containing 0.05% Tween 20 (Sigma, ref. P1379) and blocked for 1 h with a solution of 1% bovine serum albumin (BSA, Sigma, ref. A7030) in PBS. Extracellular medium samples were diluted 1:500 to 1:10,000 in the PBS-BSA solution, depending on the day of culture. The plate was washed and loaded with the diluted samples or human IgG whole molecule standards (Cederlane, ref. 009-000-003). After 1 h, the plate was washed and incubated for 1 h with peroxidase-conjugated goat anti-human Fc-specific antibodies (Sigma, ref. A0170). After washing, the plate was revealed with TMB substrate (Sigma, ref. T0440) and incubated for 20 min in the dark. Absorbance was read at 630 nm four times in the span of ten minutes and the slopes of optical density variations were compared to the standards by linear regression to determine the concentration in each well.

**Statistical analysis.** Data are presented as mean ± standard error of the mean (n = 3, except for control n = 7). Ordinary one-way or two-way (for time-dependent variables) analyses of variance (ANOVA) were performed. Values of $p < 0.05$ were considered significant and the notations of $^*$ ($p < 0.05$), $^{**}$ ($p \leq 0.01$) and $^{***}$ ($p \leq 0.001$) were used for the comparison versus the control.

## Results

### Concentration response studies for α-LA and MB

The threshold concentrations of each drug, above which CHO-DXB11 cells growth and viability are significantly affected, were first determined in 6-well plates cultures. In the case of adding α- LA (Fig 1A), the cell viability was maintained for 120 h in all cultures except for concentrations of 100 μM and above, which were stopped at 96 h when cells viability reached 90% ($> 95\%$ for the other cultures). Indeed, α-LA at 10 μM and 20 μM did not affect cell specific growth rates (average $\mu = 0.039 \pm 0.002$ h$^{-1}$) nor viability compared to the control culture. Non-significant decreases of growth rates were observed at 50 μM ($\mu = 0.037 \pm 0.002$ h$^{-1}$, $p = 0.52$) and at 100 μM ($\mu = 0.033 \pm 0.003$ h$^{-1}$, $p = 0.08$). The deleterious effects of high concentrations in α-LA were confirmed with strongly impaired growth and viability at 200 μM ($\mu = 0.029 \pm 0.002$ h$^{-1}$, $p = 0.008$) and at 500 μM ($\mu = 0.014 \pm 0.002$ h$^{-1}$, $p < 0.001$). From these results, 20 μM was selected as the highest concentration of α-LA that does not result in a detectable effect on cell growth, and 100 μM as the highest concentration that does not cause a significant decrease in cell viability.

The addition of MB showed no growth inhibition until 500 nM, with cultures at 10 nM, 100 nM and 500 nM behaving similarly to the control (average $\mu = 0.043 \pm 0.002$ h$^{-1}$, Fig 1B). At 1 μM, a minor decrease of the cells specific growth rate was observed ($\mu = 0.041 \pm 0.003$ h$^{-1}$, $p = 0.56$). Of interest, the viability was maintained at the end of the culture, with 92 ± 1% at 120 h when treated with 1 μM MB compared to 77 ± 3% for the control culture. However, the cells growth rate was significantly reduced at 10 μM ($\mu = 0.022 \pm 0.003$ h$^{-1}$, $p < 0.001$). Therefore, and following the same criterion as for α-LA, a MB concentration of 1 μM was used in the remainder of the study.

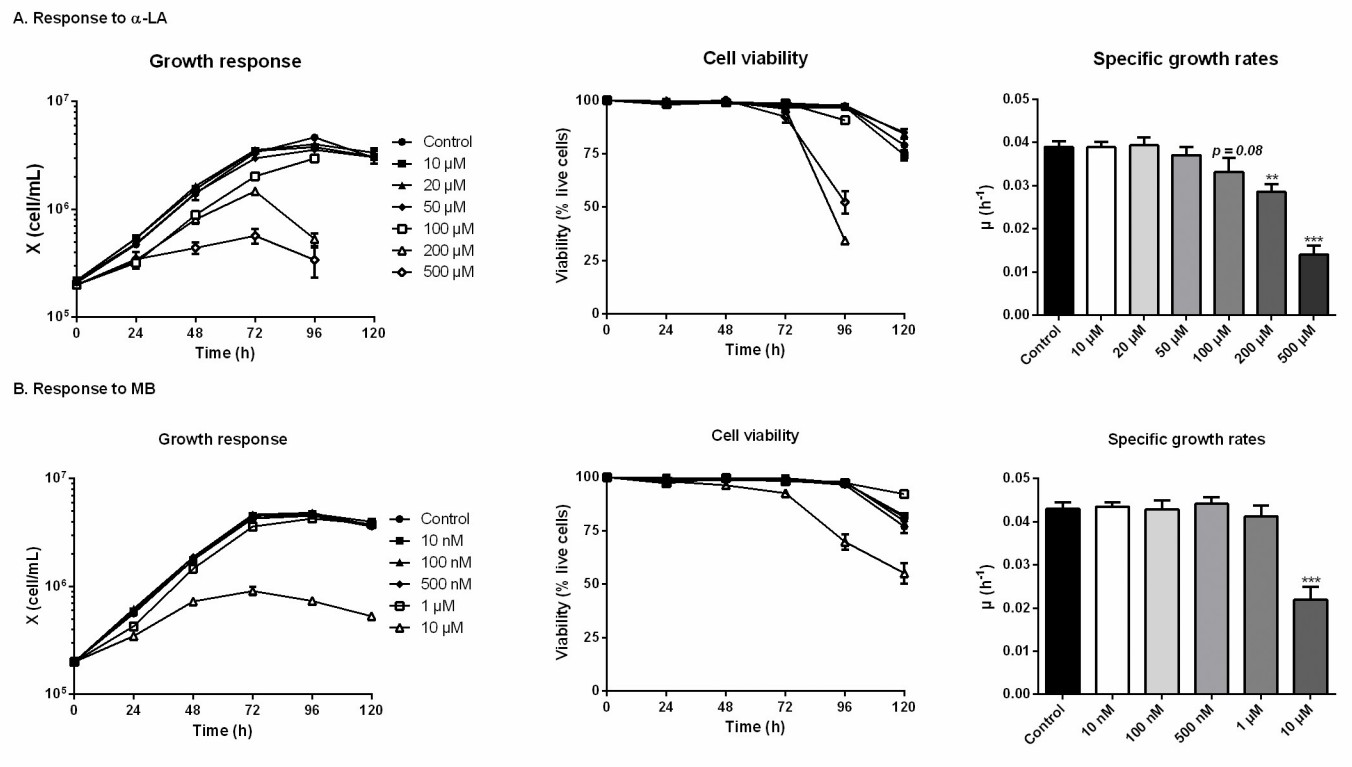

**Fig 1. Growth and viability responses of CHO cells to various doses of α-lipoic acid (A-LA) and methylene blue (MB).** α-LA was tested at 10 μM, 20 μM, 50 μM, 100 μm, 200 μM and 500 μM (A) and MB was tested at 10 nM, 100 nM, 500 nM, 1 μM and 10 μM. Growth and viability curves are presented as means ± SEM (n = 3). Specific growth rates were calculated by linear regression during the exponential growth phase, from 0 to 72 h. Statistical significance was determined by one-way ANOVA versus the control culture.

## α-LA and MB have distinct significant metabolic effects

The effect of the drugs on cell metabolism was then characterized in shake flask cultures. Drugs were assayed alone as well as combined, and compared to three controls: non-treated (control), treated only with the vehicle used for α-LA administration (0.1% ethanol, control + vehicle) and treated with a known PDH activator (DCA 5 mM). As inferred by our previous observations, similar cell growth and viability behaviors were observed in all cultures (Fig 2A), with a specific growth rate of μ = 0.040 ± 0.002 h$^{-1}$ and viability higher than 95% until 96 h, except for 100 μM α-LA where cell growth was affected (μ = 0.033 ± 0.002 h$^{-1}$, p = 0.01, Fig 2A–3). The positive impact of MB on viability was confirmed, with levels of 84 ± 1% for 1 μM MB and 82 ± 3% when combined with 20 μM A-LA at 120 h, compared to 73 ± 4% for the control (Fig 2A–2).

All drugs were added to the culture medium prior to inoculation, with the following conditions: control, 0.1% ethanol (control + vehicle), 5 mM DCA, 20 μM α-LA, 100 μM α-LA, 1 μM MB and 20 μM α-LA combined with 1 μM MB. (A) Cellular growth, viability and specific growth rates were compared to the control. (B) The glucose (GLC) consumption and lactate (LAC) production rates were compared by calculating their ratio (Y$_{LAC/GLC}$). This yield was taken from 0 to 48 h (exponential growth phase) and from 48 to 120 h (late phase), then used to quantify the glycolytic fluxes. (C) Glutamine (GLN) consumption rates were compared to glutamate (GLU) production rates before glutamine depletion (0–72 h), the resulting yield (Y$_{GLU/GLN}$) quantifies the share of glutamine directed to anaplerosis.

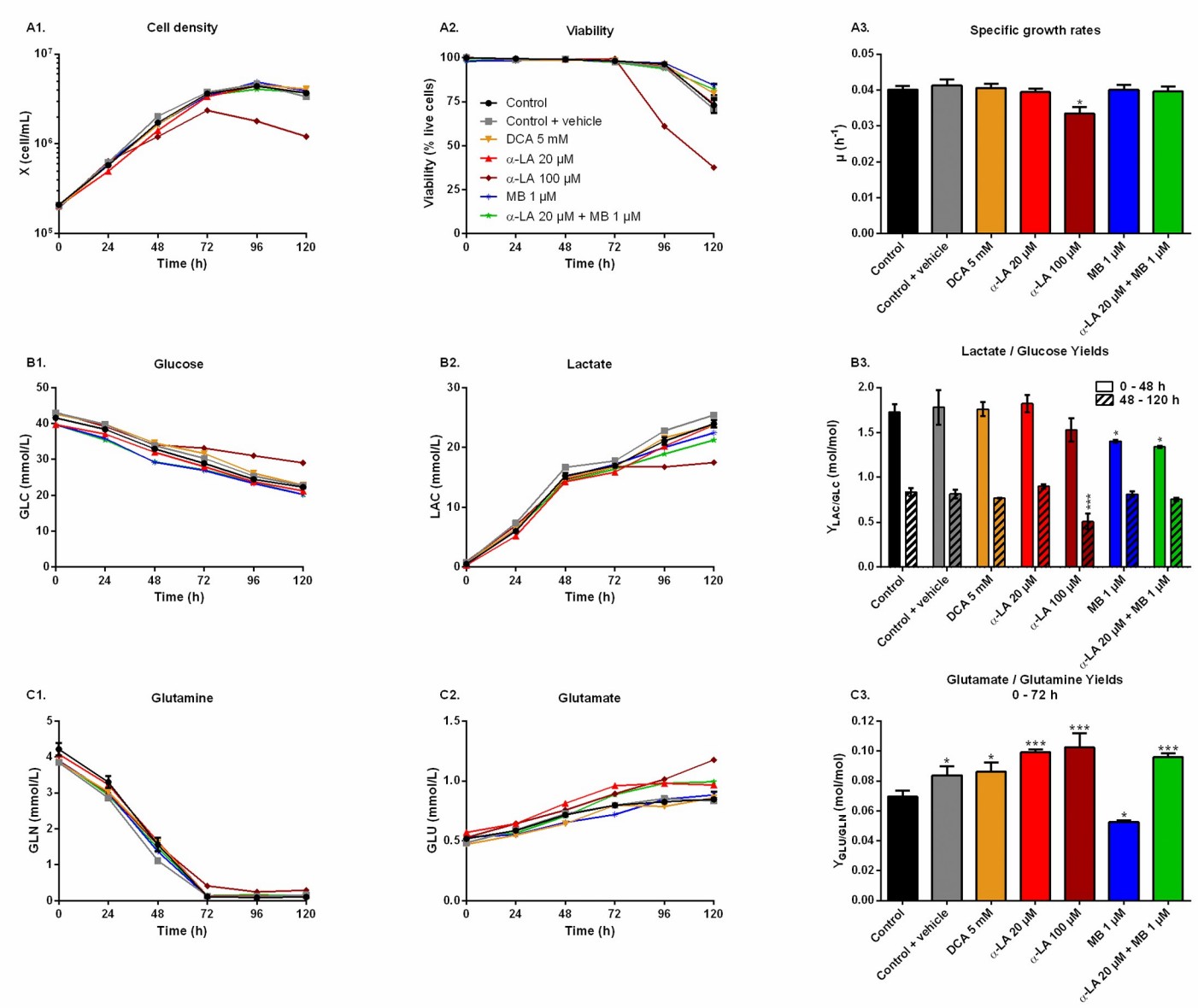

**Fig 2. Metabolic responses after drug administration.**

**Aerobic glycolysis.** Cells glycolytic metabolism was analyzed by comparing glucose consumption to lactate production (Fig 2B). Glucose specific uptake rate ($q_{GLC}$) and lactate specific production rate ($q_{LAC}$) were determined in two distinct metabolic phases, taking into account a metabolic shift observed at 48 h. The first phase was calculated from 0–48 h during the exponential growth phase, where both glucose consumption and lactate production fluxes stayed at high levels, with $q_{GLC}$ = 0.22 ± 0.01 μmol/10⁶cells/h and $q_{LAC}$ = 0.36 ± 0.02 μmol/10⁶cells/h for the control group (S1 Fig). The second phase, i.e. late growth phase (48–120 h), was characterized by lower fluxes, with a decrease of 79% for $q_{GLC}$ and 90% for $q_{LAC}$ in the control culture. Similar trends were observed in all conditions (S1 Fig). The $Y_{LAC/GLC}$ yield (-$q_{LAC}$/$q_{GLC}$) shows that in all conditions most of the uptake glucose undergoes aerobic glycolysis during exponential growth, while this phenomenon is reduced by half during the late

growth phase (Fig 2B–3). No significant differences were found when cells were treated with drug vehicle (0.1% ethanol) alone, 20 μM α-LA or its positive control 5 mM DCA. However, 100 μM α-LA resulted in a reduced contribution of aerobic glycolysis, especially during the late growth phase ($Y_{LAC/GLC}$ = 0.51 ± 0.01 mol/mol vs. 0.84 ± 0.05 mol/mol for the control). At 1 μM, MB showed to decrease $Y_{LAC/GLC}$ both alone and in combination with 20 μM α-LA for the first 48 h, with -19% and -23% versus the control, respectively, and to be similar to the control thereafter.

**Glutaminolysis.** Before glutamine depletion, observed at ~72 h in all conditions, all treatments showed strong effects on the glutaminolysis pathway, evaluated from the $Y_{GLU/GLN}$ yield (- $q_{GLU}/q_{GLN}$) at 0-72 h (Fig 2C). Glutaminolysis refers to the efficient use of glutamine, second carbon and nitrogen source, incorporated in the TCA cycle. Cells treated with 5 mM DCA showed a 24% increase of $Y_{GLU/GLN}$, and thus a decreased glutaminolysis phenomenon. However, significant $Y_{GLU/GLN}$ increases were observed at 20 μM and 100 μM α-LA, with + 43% and + 48% respectively (Fig 2C–3). It was also observed that supplementing the culture with the drug vehicle (0.1% ethanol) alone caused a slight increase of + 21% in $Y_{GLU/GLN}$, compared to control. Interestingly, the addition of MB at 1 μM showed to favor glutaminolysis with - 24% measured for $Y_{GLU/GLN}$. Finally, when used in combination with MB, the effect of α-LA was predominant with a + 38% increase in $Y_{GLU/GLN}$ (Fig 2C–3). Therefore, α-LA and DCA, both drugs known to activate the pyruvate dehydrogenase (PDH) and thus stimulate pyruvate entry into mitochondria, decreased the entry of glutamine in the TCA cycle, while MB increased glutaminolytic anaplerosis.

## Drug combination promotes cells OxPhos

The cell specific oxygen consumption rate ($q_{O2}$) observed for the control culture at 24 h, with $q_{O2}$ = 0.22 ± 0.02 $\mu molO_2/10^6$cells/h, was similar to previous data obtained with the same cell line [53]. While being maintained during exponential growth phase, $q_{O2}$ then constantly and strongly decreased (Fig 3A). Such trend was observed in both respiration and leak components of the global $q_{O2}$ (Fig 3B and 3C). The use of 5 mM DCA increased $q_{O2}$ and $q_{O2,resp}$ by up to 27% and 38% at 24 h, respectively, compared to control. However, this effect was only maintained for the growth phase, then $q_{O2}$ values decreased to control level. A concentration of 100 μM α-LA did not initially increase $q_{O2}$ but, unlike DCA, it kept the respiration level constant until 120 h (Fig 3A), with an approximate 1:1 ratio between respiration and leak (Fig 3B and 3C). No such effect was observed with α-LA at 20 μM or MB at 1 μM, although their combination allowed to partially maintain cell respiration to the end of the culture. At 120 h, combined α-LA and MB led to a $q_{O2,resp}$ value 5.6 times higher than the control (Fig 3B), with a $q_{O2,leak}$ equal to that of control (Fig 3C). Of interest, the combination of the two drugs also perturbed the distribution between leak and respiration at 24 h since, although total $q_{O2}$ remained unchanged, the leak accounted for 70% of global $q_{O2}$ instead of 50% for the control (Fig 3C).

Specific oxygen consumption rates ($q_{O2}$) were measured for the different treatments with and without the ATP-synthase inhibitor oligomycin A (1 μM) in order to determine the total $q_{O2}$ (A), its share due to leak $q_{O2,leak}$ (C) and the remaining share due to mitochondrial respiration $q_{O2,resp}$ (B). All values were normalized to the $q_{O2}$ of their control at 24 h to allow for comparison.

## Drugs affect mitochondrial membrane potential and oxidative stress level

The mitochondrial activity was assessed by FACS following two different markers: the mitochondrial membrane potential (MMP), stained by Rhodamine123, and the reactive oxygen species (ROS) generation at the membrane, stained by MitoSOX. We chose the MMP and

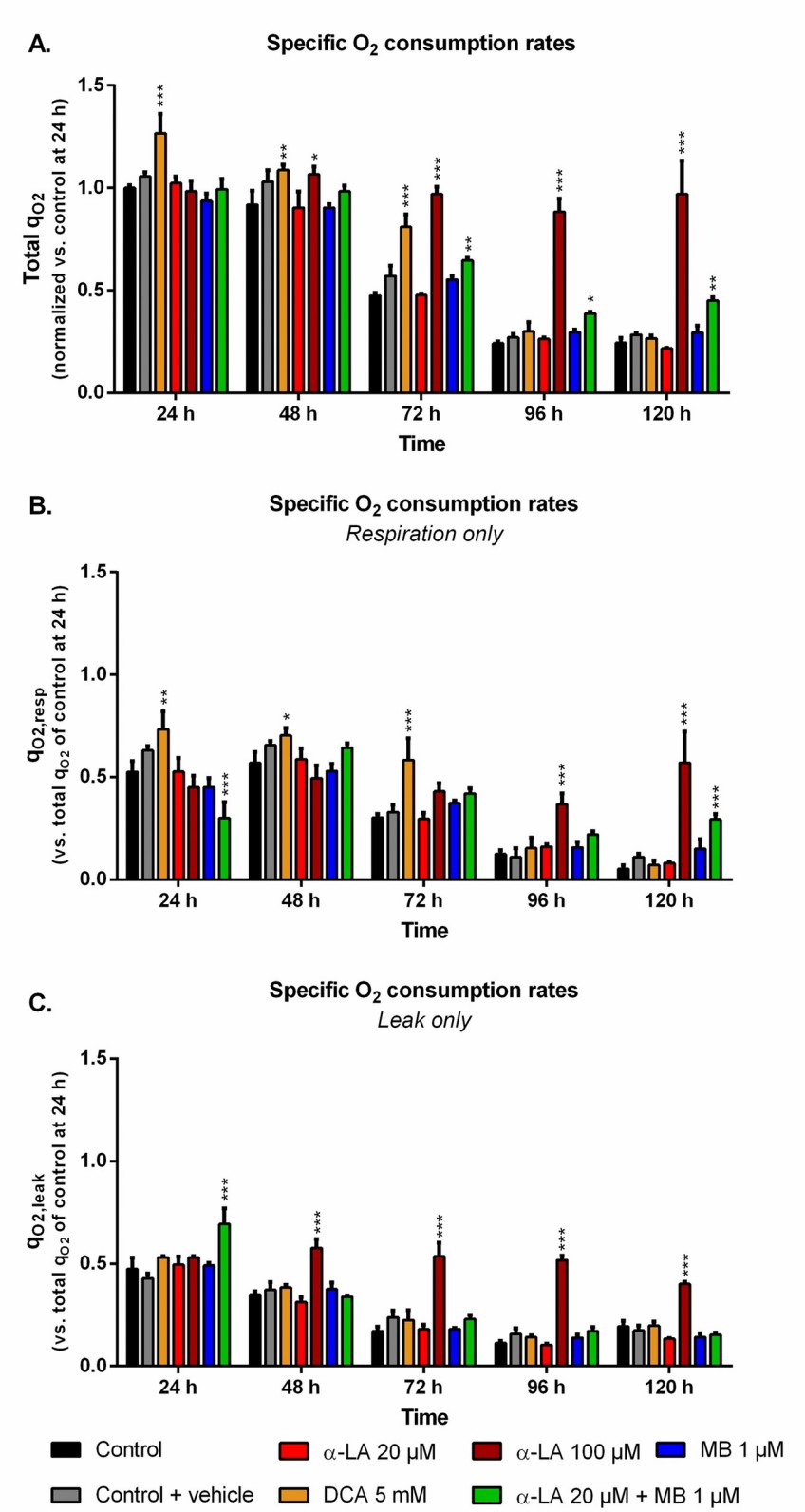

**Fig 3. Impact of the various treatments on oxygen consumption.**

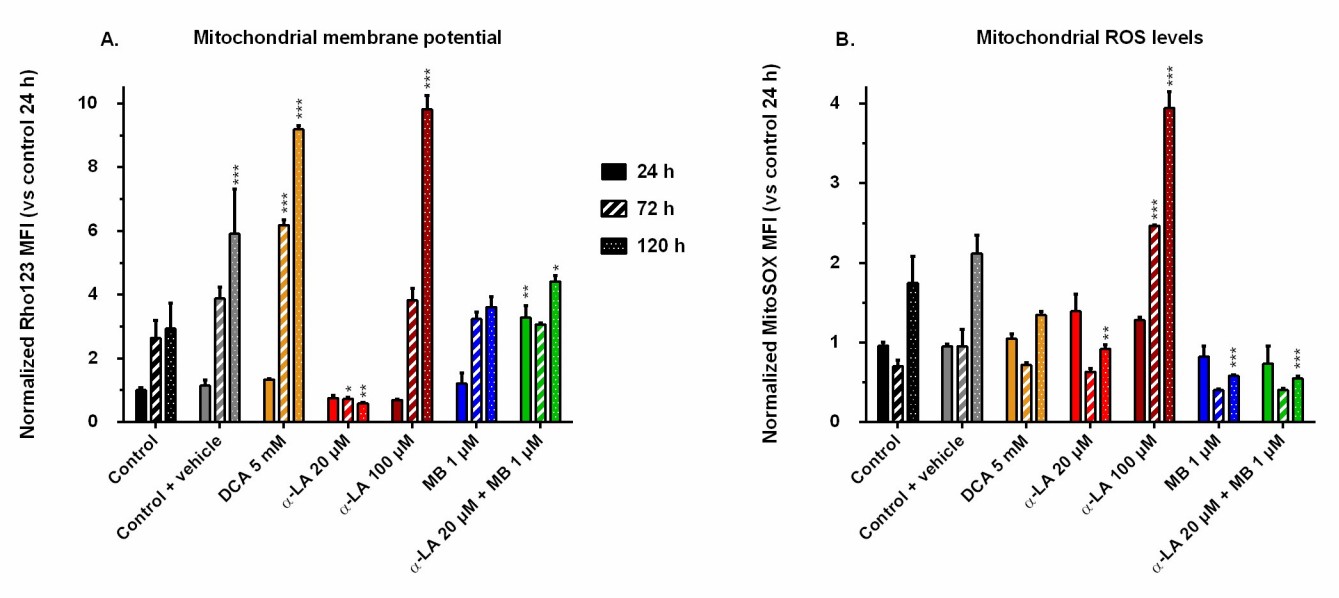

**Fig 4. Mitochondrial membrane potential and reactive oxygen species (ROS) levels induced by the drugs.** Mean fluorescence intensity (MFI) was measured by FACS after staining with Rhodamine123 (A) for the mitochondrial membrane potential, and with MitoSOX (B) for the levels of superoxide ions located at the mitochondria. All values are presented as means ± SEM with arbitrary units (normalized versus the MFI of the control at 24 h).

ROS values of control at 24 h as references for all conditions and compared their evolution to these designated references. The MMP of the control increased with time up to 3-fold after the exponential growth phase (Fig 4A), a trend opposite to that of cell respiration. The addition of 0.1% ethanol (control + vehicle) resulted in a greater but maintained MMP at the end of the culture. Pronounced increases of MMP were observed in both cultures treated with DCA at 5 mM and α-LA at 100 μM, with respective increases of 9.8 and 9.1 times the reference, measured at 120 h. In contrast, 20 μM α-LA culture maintained a low MMP, under 80% of that of the reference. Finally, the addition of MB did not affect MMP, except when combined to 20 μM α-LA where an initial burst was observed at 3.2 times the reference level, while remaining at control level until the end of the culture.

When functioning normally, the electron transport chain (ETC) generates ROS, among which superoxide ions can be stained by the MitoSOX fluorescent dye. The control, drug vehicle and 5 mM DCA (to a lesser extent) conditions showed similar trends, with stable levels at 24 and 72 h, and 1.5 to 2-fold increase at 120 h (Fig 4B). In agreement with their high mortality levels, cells treated with 100 μM of α-LA excessively generated mitochondrial ROS. Finally, at 120 h, instead of the doubling observed for the control, 20 μM α-LA, 1 μM MB and their combination showed decreasing ROS levels with respectively 0.9, 0.6 and 0.5 times the reference value (Fig 4B).

## MB significantly increases the final monoclonal antibody titer

Maximum mAb titers were reached at 96 h and decreased afterwards (Fig 5A), although not exactly following viability trends (Fig 1). A similar maximum value of 49 ± 3 mg/L was measured for the control, the 20 μM α-LA and 5 mM DCA conditions. The addition of 0.1% ethanol (control + vehicle) resulted in a final production reduction of 20% (Fig 5B). The use of 100 μM α-LA decreased the maximal titer by 67%, and it was not the result of the presence of ethanol alone (p < 0.001, one-way ANOVA vs. control + vehicle). Notably, the addition of MB

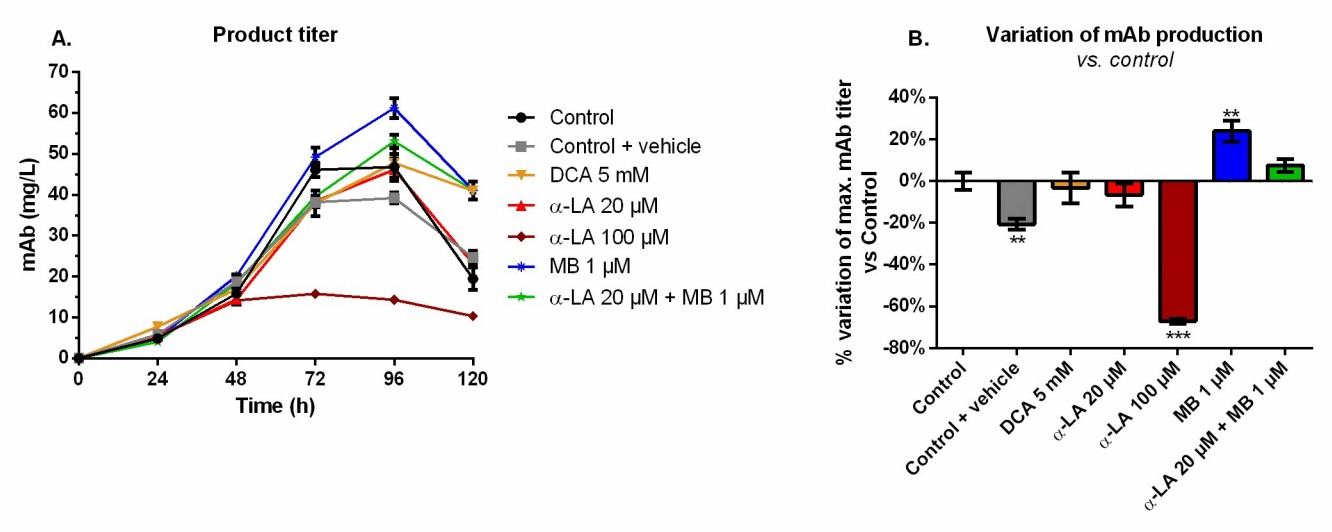

**Fig 5. Monoclonal antibody (mAb) production and variation of the maximum mAb titer in the extracellular medium for the various drug treatments versus control.** (A) Product titer was determined by ELISA and (B) its effect on mAb production is presented as the percentage of variation versus control.

at 1 μM stimulated the mAb production (+24 ± 5%, p = 0.0013). A positive but non-significant increase (+7 ± 3%, p = 0.25) was also detected when MB was combined to 20 μM α-LA.

## Discussion

### Up-regulation of pyruvate dehydrogenase can lead to OxPhos saturation

Two of the three drugs tested (i.e. α-LA and DCA) target the PDH enzyme, which converts pyruvate to mitochondrial acetyl co-enzyme A (AcCoA) rather than to extracellular lactate. However, the expected decrease of global lactate production and of $Y_{LAC/GLC}$ has only been observed at 100 μM α-LA, while it was not significant at 20 μM α-LA nor 5 mM DCA (Fig 2B). At 20 μM α-LA, the increase of $q_{LAC}$ was counterbalanced by a slight (but not significant) increase of $q_{GLC}$ (S1 Fig). Although both specific rates increased at 100 μM α-LA, the increase was higher for $q_{GLC}$ than for $q_{LAC}$, which explains a lower $Y_{LAC/GLC}$. With no significant effect on $q_{LAC}$ or $q_{GLC}$, our results with DCA differ from Buchsteiner et al. (2018), which may underly some cell line differences. However, both drugs (DCA and α-LA alone or in combination with MB) reduced glutaminolysis (Fig 2C), the main anaplerotic pathway in CHO cells [54, 55], a result mostly due to an increased $q_{GLU}$ (S1 Fig). Martínez et al. (2013) report that CHO cells maintain constant TCA fluxes by reducing glutaminolysis when other anaplerotic fluxes are activated during the glycolysis/OxPhos switch. These results suggest that α-LA and DCA-treated cells may increase their $Y_{GLU/GLN}$ ratio in order to compensate for an increased anaplerosis. Indeed, α-LA is known to activate multiple entry-point enzymes to the TCA cycle [56]. A similar conclusion was drawn by Zagari et al. (2013) who used a model of restricted mitochondrial oxidative capacity to explain the codependency of glutamine and lactate metabolisms.

Evaluating the drugs effect on mitochondrial activity homeostasis requires looking at respiratory data. In our work, the enhanced TCA activity from 5 mM DCA was confirmed by an increased total $q_{O2}$ during exponential growth (0–72 h, Fig 3). However, these increased TCA fluxes resulted, at 120 h, in a mitochondrial imbalance with proton accumulation at the

membrane (Rho123, Fig 4A) and a reduction of cellular respiration (Fig 3B). These results agree with the lower ATP concentrations at 5 mM DCA which were previously reported by Buchsteiner et al. (2018). At 100 μM α-LA, the stimulation of TCA cycle activity resulted in a maintained oxygen consumption rate from 0 to 120 h. However, as for our positive DCA control, a significant proton accumulation was observed at the mitochondrial membrane. This mitochondrial saturation at 100 μM α-LA coincided with increased levels of mitochondrial ROS (Fig 4), and proton leak flux (Fig 3C), indicating extreme levels of stress coherent with the observed decrease in cell viability. Also from using Rhodamine123 staining, Hinterkörner et al. (2007) proposed aerobic glycolysis as a mitochondrial pressure relief mechanism, which can be triggered from a high mitochondrial membrane potential. Interestingly, the addition of 20 μM α-LA did not alter the respiration and proton leak rate profiles, while maintaining low mitochondrial membrane potential and ROS levels. The mitochondrial activity and redox balance are strongly dependent on α-LA. It does not only have antioxidant properties but it also acts as cofactor of many mitochondrial enzymes in addition to its action on PDH [57]. For instance, the regulation of complex I production of superoxide anion through its interaction with 2-oxoglutarate dehydrogenase [56] can account in part for the restriction in ROS production (Fig 4). To sum up, α-LA is efficient to manage TCA replenishment and positively regulate the mitochondrial function, but at high concentration such as 100 μM and above, significant changes in mitochondrial metabolism induce damageable stress levels.

## Methylene blue enhances the mitochondrial capacity

MB at 1 μM clearly enhanced mitochondrial capacity, a conclusion supported by a coherent set of coordinated effects including lower lactate yield (i.e. more glycolytic flux to TCA cycle), higher glutaminolysis (i.e. more glutamate flux to TCA cycle), control level $q_{O2}$ and mitochondrial membrane potential, and lower ROS level. MB is a potent redox exchanger acting as an electron shuttle in the mitochondria, bypassing complexes I to III of the ETC and resulting in decreased ROS production [46, 58]. High levels of mitochondrial ROS are associated to high proton leak rates in order to dampen ROS production, thus decreasing ATP synthesis [59]. From these results, we hypothesize that 1 μM MB induces an oxidoreductive "sink" at ETC that pulls on the various anaplerotic pathways to feed the TCA cycle, explaining decreased lactate and glutamate secretion rates (S1 Fig). Such enhanced mitochondrial activity can account for the observed increase in mAb production (Fig 5). Interestingly, coupling α-LA to MB combines the effects of each drug, with a reduced aerobic glycolysis and low ROS levels. The signs of healthy mitochondria are confirmed by the significantly higher $q_{O2}$ at the end of the culture (Fig 3), although it only translated into a 7 ± 3% increase in mAb production.

## Conclusion

Our results provide further evidence on the use of metabolic approaches to understand and overcome Warburg effect-related limitations on mAb production by CHO cells. By up-regulating PDH, the α-lipoic acid (α-LA) drug proved efficient at redirecting anaplerotic fluxes towards mitochondria thus increasing TCA activity. However, α-LA above 100 μM disturbs the tightly regulated redox status at the ETC, inducing important stress signals, while 20 μM maintains a minimal stress level. Of interest, the use of methylene blue (MB) at 1 μM showed promising results with increased mitochondrial activity under minimal stress level, and increased mAb production. Although the combination of MB and α-LA led to a less pronounced increase of mAb production than using MB only, it improved cellular respiration. The coordinated actions of *pushing* on pyruvate entry into mitochondria (α-LA) and *pulling*

on anaplerotic pathways feeding the TCA cycle, while maintaining low ROS level (MB), revealed regulations that confirm the metabolic similarities between CHO and cancer cells.

At the molecular level, metabolic changes can impact mAb quality, i.e. the glycosylation profile and biological activity. Further dedicated studies would be required to identify optimal lipoic acid and methylene blue concentrations and ratios to preserve the mAb molecular properties. We chose to focus on the net production of antibody as it reflects the general metabolic state of the cell. Using this criterium, we showed that, even more than the imbalance between glycolysis and respiration, the mitochondrial capacity was critical for productivity in this CHO cell line. Altogether, the metabolic drugs originating from human therapy proved to be a convenient and efficient tool to study and direct the metabolic regulations of CHO-based bioprocesses.

## Supporting information

**S1 Fig. Specific consumption and production rates.** Specific consumption and production rates of glucose (A), lactate (B), glutamine (C) and glutamate (D) were measured in the extracellular medium for the various drug treatments. Glycolytic specific rates $q_{GLC}$ and $q_{LAC}$ were calculated on 0–48 h and 48–120 h based on the metabolic shift observed at 48 h. Glutaminolytic rates $q_{GLN}$ and $q_{GLU}$ were calculated before (0–72 h) and after (72–120 h) glutamine depletion. All conditions were statistically compared to the control by one-way ANOVA. (DOCX)

**S1 Raw data.**
(ZIP)

## Acknowledgments

Authors wish to thank Frédéric Bouillaud for helpful discussions.

## Author Contributions

**Conceptualization:** Léa Montégut, Pablo César Martínez-Basilio, Jorgelindo da Veiga Moreira, Laurent Schwartz, Mario Jolicoeur.

**Data curation:** Léa Montégut.

**Formal analysis:** Mario Jolicoeur.

**Funding acquisition:** Mario Jolicoeur.

**Investigation:** Léa Montégut, Mario Jolicoeur.

**Methodology:** Léa Montégut, Pablo César Martínez-Basilio, Mario Jolicoeur.

**Project administration:** Mario Jolicoeur.

**Supervision:** Mario Jolicoeur.

**Writing – original draft:** Léa Montégut, Mario Jolicoeur.

**Writing – review & editing:** Léa Montégut, Mario Jolicoeur.

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
