## [Decision Letter · Decision Letter 0]

24 Dec 2019

PONE-D-19-29449

Lipoic acid and methylene blue reduce the Warburg effect and enhance monoclonal antibody production in CHO cells

PLOS ONE

Dear Dr. Jolicoeur,

Thank you for submitting your manuscript to PLOS ONE. After careful consideration, we feel that it has merit but does not fully meet PLOS ONE’s publication criteria as it currently stands. Therefore, we invite you to submit a revised version of the manuscript that addresses the points raised during the review process.

Please address the comments from the editor and the reviewer. 

We would appreciate receiving your revised manuscript by 2/10/2020. To enhance the reproducibility of your results, we recommend that if applicable you deposit your laboratory protocols in protocols.io, where a protocol can be assigned its own identifier (DOI) such that it can be cited independently in the future. For instructions see: http://journals.plos.org/plosone/s/submission-guidelines#loc-laboratory-protocols

We look forward to receiving your revised manuscript.

Kind regards,

Daotai Nie, Ph.D.

Academic Editor

PLOS ONE

Journal Requirements:

Additional Editor Comments (if provided):

There are several areas that can be revised to improve the manuscript. First, lactate levels in the culture supernatants are needed to support the notion that the treatments reduce lactate production and push pyruvate into TCA cycle. Second, methylene blue enhanced mAB production, but from the data presented, lipoic acid did not enhance mAB production. This is not consistent with the claim in the title. Third, methylene blue and lipoic acid both reduce Warburg effects, why are they different in effects on mAB productions? Any other off-targets effects of the compounds?

Reviewers' comments:

Reviewer's Responses to Questions

**Comments to the Author**

1. Is the manuscript technically sound, and do the data support the conclusions?

Reviewer #1: Yes

2. Has the statistical analysis been performed appropriately and rigorously? 

Reviewer #1: Yes

3. Have the authors made all data underlying the findings in their manuscript fully available?

Reviewer #1: Yes

4. Is the manuscript presented in an intelligible fashion and written in standard English?

Reviewer #1: Yes

5. Review Comments to the Author

Reviewer #1: This is a very solid study – from background layout, to research strategy, experiment design, data collection and result presentations are all good. Results are convincing and supporting the conclusions. The mechanisms on these drugs alpha-LA and MB were also studied. I only have two comments:

(1) In addition to measure mAb titers by ELISA, the quality and integrity of assembled IgG could be tested by SDS-PAGE after IgG purification.

(2) As the IgG yields also depend on the particular mAb clone used in general. I wonder whether the drug treatment approaches studied here can be applied to different mAb clones?

6. PLOS authors have the option to publish the peer review history of their article (what does this mean?). If published, this will include your full peer review and any attached files.

Reviewer #1: Yes: Xin Ge

---

## [Author Response · Author response to Decision Letter 0]

29 Feb 2020

We would like to thank the Associate editor as well as the reviewers for their helpful and valuable comments.

Editor Comments (if provided):

There are several areas that can be revised to improve the manuscript. First, lactate levels in the culture supernatants are needed to support the notion that the treatments reduce lactate production and push pyruvate into TCA cycle.

• We agree, and Figure 2 showing metabolites concentration in culture media has been provided. A clear difference is visible only in the 100 µM ALA condition, however it is understood that specific values, accounting for the cell concentration, are more representative than these raw concentrations. 

Second, methylene blue enhanced mAB production, but from the data presented, lipoic acid did not enhance mAB production. This is not consistent with the claim in the title.

• We agree and the manuscript title has been modified to “Combining lipoic acid to methylene blue reduces the Warburg effect in CHO cells: from TCA cycle activation to enhancing monoclonal antibody production”.

Third, methylene blue and lipoic acid both reduce Warburg effects, why are they different in effects on mAB productions? Any other off-targets effects of the compounds?

• These questions are crucial to the understanding of our work, and precisions are already included in the manuscript:

o Line 344 "Indeed, α-LA is known to activate multiple entry-point enzymes to the TCA cycle [56]."

o Lines 363-368: "The mitochondrial activity and redox balance are strongly dependent on α-LA. It does not only have antioxidant properties but it also acts as cofactor of many mitochondrial enzymes in addition to its action on PDH [57]. For instance, the regulation of complex I production of superoxide anion through its interaction with 2-oxoglutarate dehydrogenase [56] can account in part for the restriction in ROS production (Fig 4)."

o Specifically concerning MB, we modified line 387: “MB is a potent redox exchanger acting as an electron shuttle in the mitochondria, bypassing complexes I to III of the ETC and resulting in decreased ROS production [46, 58].”

N.B. In supplement, Prof. Mario Jolicoeur ORCID number has been indicated: https://orcid.org/0000-0002-0875-7265

Reviewer #1: This is a very solid study – from background layout, to research strategy, experiment design, data collection and result presentations are all good. Results are convincing and supporting the conclusions. The mechanisms on these drugs alpha-LA and MB were also studied. I only have two comments:

(1) In addition to measure mAb titers by ELISA, the quality and integrity of assembled IgG could be tested by SDS-PAGE after IgG purification.

• We agree with the reviewer that mAb quality is central when addressing a bioprocess performance. However, the aim of this study was primarily to test the effect of drugs that are used in human to modulate cell energetics, can be useful to reduce the Warburg effect, which is known to limit mAb productivity in CHO cells. In order to clarify this point, we have added these sentences in the Conclusion from line 418 : “At the molecular level, metabolic changes can impact mAb quality, i.e. the glycosylation profile and biological activity. Further dedicated studies would be required to identify optimal lipoic acid and methylene blue concentrations and ratios to preserve the mAb molecular properties. We chose to focus on the net production of antibody as it reflects the general metabolic state of the cell. Using this criterium, we showed that, even more than the imbalance between glycolysis and respiration, the mitochondrial capacity was critical for productivity in this CHO cell line.”

(2) As the IgG yields also depend on the particular mAb clone used in general. I wonder whether the drug treatment approaches studied here can be applied to different mAb clones?

• We agree with the reviewer, in addition to the CHO cell line effect. It is expected that each production platform (CHO cell line – mAb expression system) will present variations of performance when assessing culture management strategies. One unexpected example is precisely the no-effect of DCA on the mAb productivity of CHOs studied in this study, conversely to an enhancement reported in other CHOs used in the study of Buchsteiner et al. 2018. Further works aiming at identifying optimal drugs’ respective concentrations and ratios on mAb productivity will definitely require scanning various CHO cell lines harboring different recombinant expression systems.

---

## [Decision Letter · Decision Letter 1]

1 Apr 2020

Combining lipoic acid to methylene blue reduces the Warburg effect in CHO cells: from TCA cycle activation to enhancing monoclonal antibody production

PONE-D-19-29449R1

Dear Dr. Dr. Jolicoeur,

We are pleased to inform you that your manuscript has been judged scientifically suitable for publication and will be formally accepted for publication once it complies with all outstanding technical requirements.

With kind regards,

Daotai Nie, Ph.D.

Academic Editor

PLOS ONE

Additional Editor Comments (optional):

Reviewers' comments:

Reviewer's Responses to Questions

**Comments to the Author**

1. If the authors have adequately addressed your comments raised in a previous round of review and you feel that this manuscript is now acceptable for publication, you may indicate that here to bypass the “Comments to the Author” section, enter your conflict of interest statement in the “Confidential to Editor” section, and submit your "Accept" recommendation.

Reviewer #1: All comments have been addressed

2. Is the manuscript technically sound, and do the data support the conclusions?

Reviewer #1: Yes

3. Has the statistical analysis been performed appropriately and rigorously? 

Reviewer #1: Yes

4. Have the authors made all data underlying the findings in their manuscript fully available?

Reviewer #1: Yes

5. Is the manuscript presented in an intelligible fashion and written in standard English?

Reviewer #1: Yes

6. Review Comments to the Author

Reviewer #1: Responses are fine. It's an interesting and solid study.

As revision reviewer, I have nothing else to say to be with more than 100 characters.

7. PLOS authors have the option to publish the peer review history of their article (what does this mean?). If published, this will include your full peer review and any attached files.

Reviewer #1: Yes: Xin Ge

---

## [Editor Report · Acceptance letter]

6 Apr 2020

PONE-D-19-29449R1 

Combining lipoic acid to methylene blue reduces the Warburg effect in CHO cells: from TCA cycle activation to enhancing monoclonal antibody production 

Dear Dr. Jolicoeur:

I am pleased to inform you that your manuscript has been deemed suitable for publication in PLOS ONE. Congratulations! Your manuscript is now with our production department. 

With kind regards,

on behalf of

Dr. Daotai Nie 

Academic Editor

PLOS ONE